# Site-wise dynamic defects in a non-conserving exclusion process

**Nikhil Bhatia⋆ and Arvind Kumar Gupta†**

Department of Mathematics, Indian Institute of Technology Ropar,
Rupnagar-140001, Punjab, India

⋆ nikhil.19maz0007@iitrpr.ac.in , † akgupta@iitrpr.ac.in

## Abstract

Motivated by the significant influence of the defects in the dynamics of the natural or man-made transportation systems, we propose an open, dynamically disordered, totally asymmetric simple exclusion process featuring bulk particle attachment and detachment. The site-wise dynamic defects might randomly emerge or vanish at any lattice location, and their presence slows down the motion of the particles. Using a mean-field approach, we obtain an analytical expression for both particle and defect density and validate them using Monte Carlo simulation. The study investigates the steady-state characteristics of the system, including phase transitions, analysis of boundary layers, and phase diagrams. Our approach streamlines the defect dynamics by integrating two parameters into one called the obstruction factor, which helps in determining an effective binding constant. The impact of the obstruction factor on the phase diagram is explored across various combinations of binding constants and detachment rates. A critical value of the obstruction factor is obtained, about which an infinitesimal change results in a substantial qualitative change in the structure of the phase diagrams. Further, the effect of the detachment rate is studied, and critical values along which the system observes a quantitative transition of the stationary phases are obtained as a function of the obstruction factor. Overall, the system shows stationary phases ranging from three to seven depending upon the value of the obstruction factor, the binding constant, and the detachment rate. Moreover, we scrutinized the impact of the obstruction factor on the shock dynamics and found no finite-size effect.

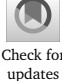

# 1 Introduction

Transport has an indispensable role in our everyday lives and over the decades, there has been a surge of interest to explore stochastic transport phenomena of various complex non-equilibrium systems ranging from natural to man-made such as vehicular traffic flow, pedestrian motion [1–5]. In eukaryotic cells, vehicles are molecular motors that proceed along intracellular filaments or DNA/mRNA strands, or ions migrating through ion channels [6–8]. One of the characteristics of all such non-equilibrium systems is a non-zero current in a steady state. In contrast to the thermodynamically balanced systems, there is no overarching theoretical framework to figure out the characteristics of the aforementioned systems. The stochastic transport in such situations is captured by the paradigmatic model totally asymmetric simple exclusion process (TASEP) [9, 10]. In its simplest incarnation, TASEP was proposed to model biopolymerization, such as the synthesis of RNA on DNA templates [11, 12]. It captures the collective non-equilibrium dynamics of active species represented by particles traveling across a one-dimensional lattice. In an open TASEP, the particles are allowed to enter and depart at the extreme ends of a lattice and hop along a preferred direction within the bulk while taking the hard-core exclusion principle into account. From a theoretical standpoint, TASEP has been extensively studied as an archetype model of jamming, helped by the property that it is exactly solvable and that a mean-field approach gives the same result as the exact solution [2, 13–20].

TASEP has undergone several generalizations that imitate different facets of transportation ranging from micro to macro. One such non-conserving model that integrates an equilibrium process, i.e., Langmuir Kinetics (LK), with the non-equilibrium process TASEP is known as TASEP-LK. The LK dynamics represent the adsorption/desorption of particles on a lattice and their rates are re-scaled while preserving the inverse proportionality to the system size in order to study the conflict between the TASEP and the LK dynamics. This model is inspired by the diffusive and directed motion along the microtubule that is alternated by the processive molecular motors [21] and encompasses several intricate aspects, including the presence of a delocalized domain wall resulting in a phase of coexistence between low and high densities [22–24].

The existence of a disorder that slows or momentarily obstructs particle movement is one of the important aspects that are visible in almost all transport systems. For instance, a vehicle on the road may be stopped or slowed down by other vehicles or periodically switching traffic lights or during gene transcription; the molecular traffic is often "roadblocked" by histones that form the core of nucleosomes or by microtubule-binding proteins, etc. [25, 26]. These obstacles (or defects) can either be static or dynamic, leading to position-dependent hopping rates (site-wise disorder) and, thus, have a significant influence on the system dynamics. The defects have been extensively studied in the context of TASEP. Earlier, TASEP with static obstacles has been studied extensively. These defects permanently reside at a location called a specific site, and these sites were assigned hopping rates that were distinct from the others. Examples include the investigation of the role of single local inhomogeneity or quenched site-wise inhomogeneity, a random distribution of spatially varying hopping rates [27–31]. Dynamic defects, on the other hand, are more pertinent to research due to their ability to replicate the dynamics of several natural and realistic transport systems. Stochastic dynamic defects, alternatively known as dynamic defects, can emerge or disappear randomly at specific sites, altering the hopping rate compared to unaffected sites. This variation may impede particle movement, but particles move at their regular hopping rate in unobstructed regions. Previously, studies have explored uncontrolled disordered systems involving a single dynamic defect that binds or unbinds at a fixed location within a TASEP model with periodic boundary conditions [32] and has also been studied for open boundary conditions [33]. Several other modifications, such as interaction dynamics [34], non-conserving dynamics [35], reservoir crowding [36], etc., were incorporated into an open TASEP model where a single dynamic defect binds/unbinds at a fixed site. Another generalization of a single dynamic defect has been proposed in a closed lattice [37] where the defect diffuses as well as binds/unbinds throughout the lattice (no fixed site).

The scenario where multiple dynamic defects appear/disappear on the lattice, also termed a site-wise disorder, has been explored less. Although it seems more realistic and is capable of mimicking natural phenomena such as the traffic jams due to the binding/unbinding of microtubule-associated proteins [38] from microtubules which are observed in several in vivo [39] and in vitro, [40] experiments. In literature, the study of site-wise disorder has been investigated under the framework of exclusion process [41, 42]. Some versions of TASEP incorporating dynamic disorder (ddTASEP) have been investigated in a resource-constrained environment [43] whereas in [44], the model has additional feedback (the particle-defect interaction) where defects are removed by particles. Further, recently an effort has been made to numerically study a generalization of an open ddTASEP model that incorporates the Langmuir kinetics for particles [45]. However, it lacks three crucial aspects: (i) the role of defects in the particle dynamics is not incorporated at the boundary sites which ultimately govern the stationary properties of the system such as boundary-induced phase transitions; (ii) lack of uniform proportionality in the affected attachment rate and affected hopping rate of particles due to defects and (iii) the steady-state numerical solution for density is insufficient to characterize the influence of all the parameters. Therefore, in light of the above-mentioned shortcomings, we propose to analyze the role of the non-conserving dynamics of the totally asymmetric simple exclusion process with the dynamic disorder. In contrast to the previously studied model [42, 45], we have inculcated the concept of an affected hopping rate at the entry site also, which significantly impacts the system's stationary state properties, particularly when compared to the reference [45], the obstruction due to the defects in the proposed model affects the particle hopping and attachment rates in uniform proportion. To explore the dynamics of the model, we approach the system theoretically via mean-field approximation, and we mainly intend to address the following points: (i) What impact do site-wise dynamic defects have on the stationary properties of the standard TASEP-LK system, including particle

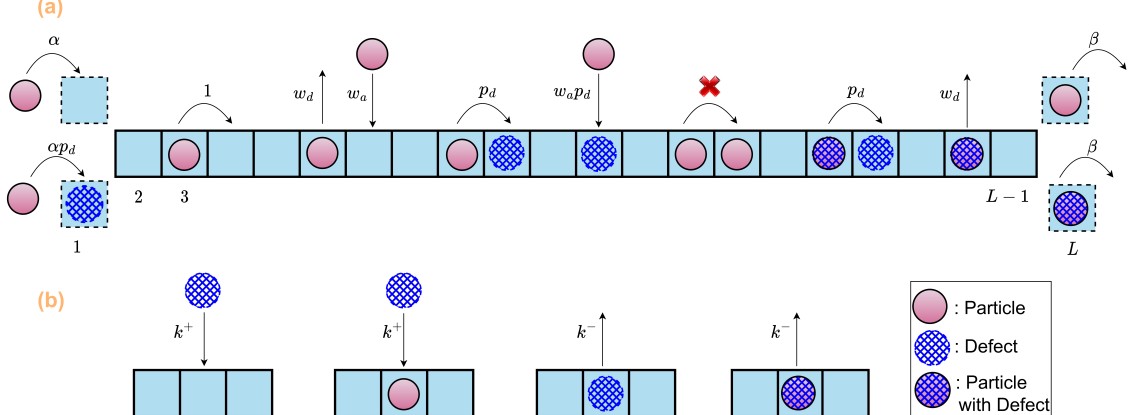

Figure 1: A diagrammatic representation of a non-conserving TASEP model depicting the dynamics of particles (highlighted in pink) and defects or obstacles (illustrated in a blue mesh). (a) Depicts different particle dynamics, including entry, exit, hopping, and attachment/detachment, along with the corresponding rates at which these events occur in the presence and absence of defects. (b) Illustrates the dynamics of defects on the lattice, including defect binding/unbinding and their corresponding rates.

flux, density profiles, and stationary phases? (ii) What factors affect the system's stationary properties? (iii) Does the system remain symmetric with respect to particle-hole in the presence of dynamic defects? (iv) Does the domain wall remain localized in the presence of defects? If yes, what is the impact of defects on the domain wall?

## 2 Model overview

In actual transportation scenarios, obstacles frequently impede movement. On highways, these obstacles might be intersections or traffic signals, while in the microscopic domain, molecular traffic is often obstructed by proteins that are bound or temporary alterations to the 'lanes' through which traffic flows. Motivated by these stochastic disorders, we propose a model representing an open, dynamically disordered TASEP with LK dynamics. It is represented through a one-dimensional discrete lattice comprising $L$ sites, each labeled from $j = 1$ to $L$. Here, particles enter through the initial site ($j = 1$), traverse both horizontally and vertically within the bulk ($2 \leq j \leq L - 1$), and exit via the final site ($j = L$). Particles moving horizontally only exhibit a unidirectional horizontal movement (left to right). Moreover, the adsorption/ desorption of particles also pertains to the lattice, where particles can also join or leave the lattice by a vertical movement from all sites other than the first and last sites. The lattice also includes a different type of entity known as defects (or obstacles), which introduce dynamic disorder and impede the movement of particles throughout the lattice. In contrast to particle movement, the defects only exhibit vertical movement and can randomly bind/unbind on every lattice site. Individually, both particles and defects adhere to the hard-core exclusion principle. Therefore, each site can only accommodate a single particle, defect, or a combination of the two. As depicted in FIG. 1, the events showcasing possible particles and defects dynamics on the lattice, along with their corresponding occurrence probabilities, are illustrated as follows:

1. **Particle dynamics**: The dynamics of particles are significantly influenced by defect occupancy; hence, these dynamics at various lattice locations are characterized as follows:

   (a) **At entry:** If the first site has no particle, a particle can enter the lattice through this site with a rate $\alpha$ if it has no defect or with a rate $\alpha p_d$ ($p_d < 1$) otherwise. In case the first site is particle-occupied and its immediate right neighbor is particle-vacant, the particle can move to this neighboring site at a unit rate if the arrival site has no defect or with a rate of $p_d$ otherwise.

   (b) **At bulk:** If a particle occupies a bulk site, it first attempts detachment at a rate of $w_d$. If detachment fails and its immediate right neighbor is particle-vacant, the particle moves to the neighboring site with a unit rate if no defect is present at the arrival site or with a rate of $p_d$ otherwise. At a bulk site without a particle, a particle can attach at a rate of $w_a$ if no defect is present at the arrival site or at a rate of $w_a p_d$ otherwise.

   (c) **At exit:** A particle present at the last lattice site can leave the lattice with a rate $\beta$.

2. **Defect dynamics**: A defect can randomly bind (unbind) at a site without (with) a defect with a rate $k^+$ ($k^-$). Note that the particle's presence on the arrival site has no effect on the dynamics of defects, but the converse is not true.

An event such as hopping of the particle, attachment/detachment of the particle, or binding/unbinding of the defect is selected depending on the probability proportional to their corresponding rates.

Notably, the proposed model is distinctive from the ref. [42] in the sense that the attachment and detachment of particles are considered to make it more realistic. Moreover, this study not only addresses the dynamics of defects (binding/unbinding) at the boundary sites but also examines its impact on particle dynamics through modified rates at the boundaries, a consideration which was absent in the references [42, 45]. In the later part, we will explicitly discuss that these considerations will produce a non-trivial effect on the stationary-state characteristics of the model. In the subsequent section, we will offer mathematical underpinning by formulating master equations that depict the temporal evolution of the average particle and defect density, elucidating the process involved, and obtaining the stationary-state solution by solving them in the thermodynamic limit.

## 3 Master equations

Individually, both particles and defects obey the hard-core exclusion principle; therefore, we introduce two binary random variables $\sigma_j$ and $\nu_j$ each denoting the occupancy of the particle and defect on the lattice, respectively. The random variable $\sigma_j$ (or $\nu_j$) $= 0/1$ signifies the absence/presence of particle (or defect) at $j^{\text{th}}$ lattice site. Now, these variables are employed to formulate the master equation showcasing the evolution of the average occupation number for each entity, starting with the particles. The particle density in the bulk of the lattice evolves as follows:

$$\frac{d\langle \sigma_j \rangle}{dt} = J_{j-1,j} + w_a \langle (1-\nu_j)(1-\sigma_j) \rangle + w_a p_d \langle \nu_j (1-\sigma_j) \rangle - J_{j,j+1} - w_d \langle \sigma_j \rangle, \qquad (1)$$

where

$$J_{j-1,j} = \langle \sigma_{j-1}(1-\nu_j)(1-\sigma_j) \rangle + p_d \langle \sigma_{j-1}\nu_j(1-\sigma_j) \rangle, \qquad (2)$$

$\langle \cdots \rangle$ denotes the statistical average and $J_{j-1,j}$ is the particle-flux from $j-1^{\text{th}}$ site to $j^{\text{th}}$ site. The equation governing the evolution of particle density at both the left-lattice and right-lattice boundaries is formulated as:

$$\frac{d\langle \sigma_1 \rangle}{dt} = \alpha \langle (1-\sigma_1)(1-v_1) \rangle + \alpha p_d \langle (1-\sigma_1)v_1 \rangle - J_{1,2}, \qquad \text{and} \tag{3}$$

$$\frac{d\langle \sigma_L \rangle}{dt} = J_{L-1,L} - \beta \langle \sigma_L \rangle, \tag{4}$$

respectively. Lastly, the master equation dictating the evolution of the average defect density within the lattice is provided as follows:

$$\frac{d\langle v_j \rangle}{dt} = k^+ \langle 1 - v_j \rangle - k^- \langle v_j \rangle, \qquad 1 \le j \le L. \tag{5}$$

In order to comprehend the stationary-state dynamics of the system, the aforementioned equations require a solution. However, solving them in their current state poses a challenge due to the presence of one-, two-, and three-point correlators. Therefore, in the subsequent section, mean-field approximations are applied to these equations in an attempt to elucidate stationary-state attributes such as density profile, potential stationary phases, phase transitions, and current.

## 4 Continuum mean-field approximations

By employing mean-field approximations, all potential particle-particle and particle-defect correlations are disregarded within the aforementioned system of master equations, namely $\langle \sigma_j \sigma_{j+1} \rangle = \langle \sigma_j \rangle \langle \sigma_{j+1} \rangle$ and $\langle \sigma_j v_{j+1} \rangle = \langle \sigma_j \rangle \langle v_{j+1} \rangle$. Additionally, we introduce the notations $\rho_j = \langle \sigma_j \rangle$ and $\rho_{d,j} = \langle v_j \rangle$ to represent the average particle density and defect density, respectively, at site $j$. This simplification results in reducing Eq. (1) to:

$$\frac{d\rho_j}{dt} = J_{j-1,j} + w_a \big( (1-\rho_{d,j})(1-\rho_j) + p_d \rho_{d,j}(1-\rho_j) \big) - J_{j,j+1} - w_d \rho_j, \tag{6}$$

where

$$J_{j-1,j} = \rho_{j-1}(1-\rho_j)(1-\rho_{d,j} + p_d \rho_{d,j}). \tag{7}$$

The evolution equations for average particle density at the left and right boundaries are reformulated as:

$$\frac{d\rho_1}{dt} = \alpha(1-\rho_1)(1-\rho_{d,1} + p_d \rho_{d,1}) - J_{1,2}, \qquad \text{and} \tag{8}$$

$$\frac{d\rho_L}{dt} = J_{L-1,L} - \beta \rho_L, \tag{9}$$

respectively. Furthermore, the evolution of average defect density within the lattice follows the subsequent equation:

$$\frac{d\rho_{d,j}}{dt} = k^+(1-\rho_{d,j}) - k^- \rho_{d,j}, \qquad 1 \le j \le L. \tag{10}$$

To obtain the continuum version of the model, we coarse-grain the lattice by introducing $x = \epsilon j \in [0,1]$ as the quasi-continuous space variable and $\epsilon = \frac{1}{L}$ as the lattice constant. Then, the terms up to the first order of $\epsilon$ are retained in the Taylor series expansion of $\rho_{j\pm1} \approx \rho(x\pm\epsilon)$ in Eq. (6) to get the reformulation of Eq. (6) and Eq. (10) as:

$$\frac{\partial \rho}{\partial t'} + \frac{\partial J}{\partial x} = \Omega_a(1 - \rho_d + p_d\rho_d)(1-\rho) - \Omega_d\rho, \tag{11a}$$

$$\frac{\partial \rho_d}{\partial t} = k^+(1-\rho_d) - k^-\rho_d, \tag{11b}$$

respectively. Here, $t' = \frac{t}{L}$ is the re-scaled time variable, and $\Omega_a = w_a L, \Omega_d = w_d L$ are the modified Langmuir kinetic rates. Furthermore, the subscript $j$ is also omitted, considering the spatial homogeneity of the lattice.

It is essential to utilize a modified detachment rate that is constant for $L \to \infty$ (in large systems) because the discrepancy between bulk and boundary dynamics becomes apparent only if particles remain on the lattice for a sufficient duration before detachment. A similar rationale justifies the adjusted attachment rate. The average particle current within the lattice bulk, considering a finite $\epsilon$, is expressed as $J = (1-\rho_d+p_d\rho_d)\left(-\frac{\epsilon}{2}\frac{\partial \rho}{\partial x} + \rho(1-\rho)\right)$ whereas in the thermodynamic limit ($\epsilon \to 0^+$), it becomes $J = (1-\rho_d+p_d\rho_d)\rho(1-\rho)$. The right-hand side of the Eq. (11a) can also be expressed as $\Omega_d(K^*+1)\left(\frac{K^*}{K^*+1} - \rho\right)$. This suggests that the density governed by the Langmuir isotherm ($\rho_l$) defined as $\frac{K^*}{K^*+1}$ will exhibit either an attracting or a repelling behavior with respect to the nonlinear relationship between current and density because this net source term is positive or negative, depending on whether the density $\rho$ is below or above $\rho_l$ where $K^* = K(1-\rho_d+p_d\rho_d)$ and $K = \frac{\Omega_a}{\Omega_d}$ is the binding constant. This will prove to be a crucial concept while discussing density profiles in subsequent sections. If the density at the left end dips below the Langmuir isotherm and the current-density relation's slope is positive, $\frac{\partial J}{\partial \rho} > 0$, then the particles will accumulate into the bulk of the lattice through Langmuir kinetics. Consequently, the density increases towards $\rho_l$ as one progresses away from the left boundary. Conversely, with a negative slope ($\frac{\partial J}{\partial \rho} < 0$), indicating densities greater than 1/2, the density profiles diverge from the Langmuir isotherm as one moves away from the left boundary [24].

The hindrance to particle movement within the lattice is directly proportional to the number of defects present on the lattice, or equivalently $\rho_d$, and inversely proportional to the affected hopping rate $p_d$. Consequently, we have introduced an obstruction factor that rationalizes the role of defects in impeding particle movement and reduces the model's parameter space. This simplification will facilitate the focused study of defects on the stationary-state characteristics of the system in subsequent sections. It is defined as:

$$z = \rho_d(1-p_d). \tag{12}$$

Utilizing the Eq. (12), the expression for the stationary-state current in the bulk of the lattice reduces to:

$$J = (1-z)\rho(1-\rho). \tag{13}$$

The above-obtained expression for the particle current indicates that the proposed model can be perceived as a generalization of the standard TASEP model or a model with static localized defects, where the effective hopping rate of particles is $1-z$ [32]. Note that the obstruction factor, being a function of $\rho_d$ and $p_d$, remains confined within the range of 0 and 1, as both parameters are bounded in the same range. The obstruction on the lattice diminishes to zero either when there are no defects on the lattice ($\rho_d = 0$) or when the affected hopping rate due to defects attains the standard unit hopping rate ($p_d = 1$). For this case, the expression for the current in Eq. (13) shows that the model reduces to that of a standard open TASEP with

LK dynamics [24]. Conversely, the particle faces maximum hindrance when all lattice sites are entirely occupied by defects, i.e., $\rho_d = 1$, and simultaneously, the defects prevent particle hopping in their presence, indicated by $p_d = 0$. For this case, the particle current vanishes from the lattice and can be easily validated from Eq. (13).

In the next section, we will obtain a stationary state analytical solution to the derived continuum equations for the particle as well as defect density and compare it to simulation results.

# 5 Analytical solution at stationary state

Theoretical defect density at the stationary state can be readily computed from Eq. (11b) as:

$$\rho_d = \frac{k^+}{k^+ + k^-}. \tag{14}$$

At stationary state, the nonlinear differential Eq. (11a) in the limit $\epsilon \to 0$, reduces to a first order differential equation,

$$\frac{\partial J}{\partial x} = \Omega_d(K^* + 1)\left(\frac{K^*}{K^* + 1} - \rho\right). \tag{15}$$

Next, we will elucidate in detail how one can analytically solve the continuum equation, Eq. (11a), in the steady state. This discussion will lead to a categorization of the potential solutions based on the entry rate ($\alpha$), exit rate ($\beta$), the effective binding constant ($K^* = K(1-z)$), and the detachment rate ($\Omega_d$).

One can easily verify that the Eq. (11a) of the system remains invariant under the following transformations: $\rho(x) \leftrightarrow 1 - \rho(1 - x)$, $w_a(1-z) \leftrightarrow w_d$. This implies that $K^* \leftrightarrow 1/K^*$ and hence this symmetry with respect to $K^*$ allows us to restrict our choices to values with $K^* \geq 1$. Then, the two scenarios that need to be distinguished are $K^* = 1$ and $K^* > 1$. The scenario where $K^* = 1$ is somewhat hypothetical and requires careful manipulation of the binding constant and obstruction factor, but it is technically more straightforward to analyze. Therefore, we will address this case first. Additionally, we will compare these results with the outcomes obtained from Monte Carlo simulations.

## 5.1 Analysis for $K^* = 1$

Theoretical computation of the average particle density becomes mathematically simplified when $K^* = 1$, as Eq. (15) factorizes to:

$$(2\rho - 1)\left((1-z)\frac{\partial \rho}{\partial x} - \Omega_d\right) = 0. \tag{16}$$

Upon solving Eq. (16), we retrieve two different solutions: a constant density $\rho_{\text{MC}}(x) = \frac{1}{2}$ associated with a maximal-current (MC) phase, and a linear profile $\rho(x) = \frac{\Omega_d}{1-z}x + C$. These solutions are similar to the case of TASEP-LK without dynamic defects [24] except for the normalization of the coefficient of $x$ in the linear solution. To ascertain the value of the integration constant $C$ in the linear density profile, we first determine the estimate to boundary densities $\rho_1$ and $\rho_L$ utilizing Eqs. (8) and (9) as:

$$\rho_1 = \alpha, \quad \text{and} \quad \rho_L = 1 - \beta^*, \tag{17}$$

where $\beta^* = \frac{\beta}{1-z}$. Now, the linear density profile yields two solutions: an entry-dominated one, corresponding to the low-density (LD) phase, achieved by matching the linear solution

with the left boundary; and another exit-dominated, corresponding to the high-density (HD) phase, obtained by matching the linear solution with the right boundary. These solutions are as follows:

$$\rho_\alpha(x) = \frac{\Omega_d}{1-z}x + \alpha,$$
$$\rho_\beta(x) = \frac{\Omega_d}{1-z}(x-1) + 1 - \beta^*. \tag{18}$$

Since we have the density solution for the standard stationary phases, we can derive a general density profile $\rho(x)$ by combining three possible solutions: $\rho_\alpha$, $\rho_\beta$, and $\rho_{MC}$. Firstly, the position separating the low-density profile $\rho_\alpha(x)$ from the density profile $\rho_{MC}(x)$ is computed as $x_\alpha = \frac{(1-2\alpha)(1-z)}{2\Omega_d}$. Additionally, we compute the position $x_\beta = \frac{2\beta + 2\Omega_d + z - 1}{2\Omega_d}$ that separates the high-density profile $\rho_\beta(x)$ from the density profile $\rho_{MC}(x)$. Depending on the relative ordering of the $x_\alpha$ and $x_\beta$, the density profiles are obtained as follows: Various scenarios arise depending on the relative ordering of $x_\alpha$ and $x_\beta$, and the corresponding density profiles for these situations are provided as follows:

1. If $x_\alpha \leq x_\beta$, the continuous and piecewise linear density profile exhibiting the co-existence of three phases is given by:

$$\rho(x) = \begin{cases} \frac{\Omega_d}{1-z}x + \alpha, & 0 \leq x \leq x_\alpha, \\ \frac{1}{2}, & x_\alpha \leq x \leq x_\beta, \\ \frac{\Omega_d}{1-z}(x-1) + 1 - \beta^*, & x_\beta \leq x \leq 1. \end{cases} \tag{19}$$

2. If $x_\alpha > x_\beta$, a jump discontinuity between the densities $\rho_\alpha(x)$ and $\rho_\beta(x)$, arises at a point $x_w$ in the form of a shock. The density profile exhibiting the co-existence of two phases is given by:

$$\rho(x) = \begin{cases} \frac{\Omega_d}{1-z}x + \alpha, & 0 \leq x \leq x_w, \\ \frac{\Omega_d}{1-z}(x-1) + 1 - \beta^*, & x_w \leq x \leq 1, \end{cases} \tag{20}$$

where the position of the shock $x_w = \frac{\beta - \alpha(1-z) + \Omega_d}{2\Omega_d}$ is obtained by utilizing the current-continuity principle at the discontinuity $x_w$. For $x_w \in (0,1)$, the shock is to be visible in the bulk of the lattice. Moreover, for $x_w \leq 0$ ($x_w \geq 1$), the shock or the LD-HD co-existence phase exits from the left (right) end of the lattice leading to the LD (HD) phase whose density profile is given by $\rho_\beta(x)$ ($\rho_\alpha(x)$). The height of the shock $\Delta$ is given by,

$$\Delta = \rho_\beta(x_w) - \rho_\alpha(x_w) = 1 - (\alpha + \beta) - \frac{\Omega_d}{1-z}. \tag{21}$$

In the limit $z \to 0$, all the above-obtained results match that of an open TASEP with LK [24] whereas in the limit $\Omega_d \to 0^+$, the LK dynamics begin to vanish from the lattice and the stationary state density profiles converge to that of an open TASEP with site-wise dynamic defects [43].

### 5.1.1 Existence of stationary phases

We briefly review the stationary properties of the homogeneous open TASEP, extensively studied through mean-field analysis. It was observed that the system could exist in one of three phases depending on the entry and exit rates: entry-dominated low density (LD), exit-dominated high density (HD), and bulk-dominated maximal current (MC). The transition from both LD and HD phases to the MC phase occurs as a second-order transition concerning density.

However, the phase transition from LD to HD is first-order. In this regard, when the entry rate equals the exit rate, an LD-HD coexistence phase (Shock (S) phase) emerges, characterized by a delocalized shock traversing the lattice. Upon the incorporation of Langmuir Kinetics, the shock becomes anchored (localized shock) and extends beyond a line, encompassing a region. Furthermore, we observe various combinations of the primary phases LD, MC, and HD [23, 24].

In our proposed model, the lattice can possess a maximum of 21 different combinations of key phases LD, HD, and MC. However, not all of them may exist for any parameter value. Now we discuss in detail the existence of the probable stationary phases and theoretically derive their existential conditions.

(a) **LD phase:** In a lattice within an entry-dominated phase, the density profile is delineated by $\rho_\alpha(x)$ with a boundary layer on the right end. The phase boundaries containing the LD phase in the $\alpha - \beta$ parameter space are specified as:

$$\alpha < \min\left(\beta - \Omega_d, \frac{1-z}{2} - \Omega_d\right). \tag{22}$$

(b) **HD phase:** In a lattice characterized by an exit-dominated phase, the density profile is given by $\rho_\beta(x)$, with a boundary layer present at the left end. The phase boundaries encompassing the HD phase within the $\alpha - \beta$ parameter space are outlined as follows:

$$\beta < \min\left(\alpha(1-z) - \Omega_d, \frac{1-z}{2} - \Omega_d\right). \tag{23}$$

(c) **MC phase:** Following the expression of the current, the gradient of the current vanishes, and the maximal current is attained for $\rho = 1/2$. Hence, in this phase, the density profile in the bulk of the lattice is given by $\rho_{MC}(x) = 1/2$, along with the presence of boundary layers at both ends. This phase exists when $\alpha$ and $\beta^*$ satisfies:

$$\alpha > \frac{1}{2}, \quad \text{and} \quad \beta^* > \frac{1}{2}. \tag{24}$$

(d) **S phase:** In the shock phase, the density profile consists of a curve that is discontinuous at a point $x_w$, combining low and high-density profiles. The density to the left of $x_w$ is represented by $\rho_\alpha(x)$, and to the right of $x_w$, it is denoted by $\rho_\beta(x)$. The conditions for the presence of this phase in the lattice are as follows:

$$\beta + \alpha(1-z) < 1 - z - \Omega_d, \quad \text{and} \quad |\beta - \alpha(1-z)| < \Omega_d. \tag{25}$$

(e) **LD-MC phase:** There exists a two-phase co-existence region (or LD-MC phase) wherein the density at the left of $x_\alpha$ is expressed by $\rho_\alpha(x)$ and at the right of $x_\alpha$ is given by $1/2$ with a boundary layer on the right end. The conditions for the existence of this phase in the lattice are given as:

$$\frac{1}{2} - \frac{\Omega_d}{1-z} < \alpha < \frac{1}{2}, \quad \text{and} \quad \beta^* > 1/2. \tag{26}$$

(f) **MC-HD phase:** The density profile for the two-phase coexisting region (or MC-HD phase) is given by a continuous combination of two curves. To the left of $x_\beta$, the density is $1/2$, while to the right of $x_\beta$, it is represented by $\rho_\beta(x)$ with a boundary layer on the left end. The conditions for the presence of this phase in the lattice are outlined as follows:

$$\alpha > \frac{1}{2}, \quad \text{and} \quad \frac{1-z}{2} - \Omega_d < \beta < \frac{1-z}{2}. \tag{27}$$

(g) **LD-MC-HD phase:** Similarly, a three-phase coexistence region (or LD-MC-HD phase) may occur. As mentioned earlier, it exists when $x_\alpha \leq x_\beta$, and the condition for its presence in the lattice is given by:

$$\beta + \alpha(1-z) > 1 - z - \Omega_d, \quad \alpha < \frac{1}{2}, \quad \text{and} \quad \beta^* < \frac{1}{2}. \tag{28}$$

Now, we provide the argument to discard the prospect of the existence of the remaining fourteen phases. The existence of the three phases MC-LD, HD-LD, HD-MC can be discarded based on the argument that it is impossible to concatenate the density profiles for the above-discussed phases either continuously or discontinuously for $\Omega_d > 0$ while keeping $\rho_\alpha(x) < 1/2, \rho_\beta(x) > 1/2$. The rest eleven co-existing three phases involve the combination with any of the above three discarded phases and hence can be discarded following a similar argument. For example, the LD-MC-LD ceases to exist because it is a combination of the LD phase with the MC-LD phase, and the latter has already ceased to exist. Therefore, up to seven distinct stationary phases may be observed in the phase diagram when $K^* = 1$.

## 5.2 $K^* \neq 1$

Considering the particle-hole symmetry, we restrict our focus to the case $K^* > 1$. In contrast to the previous case i.e., $K^* = 1$, the continuum equation governing the particle density in Eq. (15) cannot be simplified, rendering the analysis considerably more intricate. For additional analysis, we transform Eq. (15) into the format of a re-scaled density $\sigma$, for which the solution is already established [24]:

$$\sigma(x) = \frac{K^* + 1}{K^* - 1}(2\rho - 1) - 1. \tag{29}$$

Clearly, the density $\rho(x) \in [0, 1]$ implies that the re-scaled density $\sigma(x) \in \left[\frac{-2K^*}{K^*-1}, \frac{2}{K^*-1}\right]$ and here the condition $\sigma(x) = 0$ represents the Langmuir isotherm $\rho_l = \frac{K^*}{K^*+1}$ which is similar to that in [24]. The continuum equation (15) simplifies to:

$$\left(\frac{\sigma + 1}{\sigma}\right) \frac{\partial \sigma}{\partial x} = \frac{(K^* + 1)^2 \Omega_d}{(K^* - 1)(1 - z)}. \tag{30}$$

Integrating the aforementioned equation results in:

$$|\sigma(x)| \exp(\sigma(x)) = Y(x), \tag{31}$$

where $Y(x)$ is given by:

$$Y(x) = |\sigma(x_0)| \exp\left(\frac{(K^* + 1)^2 \Omega_d}{(K^* - 1)(1 - z)}(x - x_0) + \sigma(x_0)\right), \tag{32}$$

and $x_0$ is a reference point that takes on the value of 0 or 1, as the values of $\sigma(x_0)$ are known at the boundaries, thus providing:

$$\begin{aligned} Y_\alpha(x) &= |\sigma(0)| \exp\left(\frac{(K^* + 1)^2 \Omega_d}{(K^* - 1)(1 - z)}x + \sigma(0)\right), \\ Y_\beta(x) &= |\sigma(1)| \exp\left(\frac{(K^* + 1)^2 \Omega_d}{(K^* - 1)(1 - z)}(x - 1) + \sigma(1)\right). \end{aligned} \tag{33}$$

Equation (31) possesses an explicit solution expressed in terms of the Lambert-$W$ function, and can be formulated as:

$$\begin{aligned} \sigma(x) &= W(Y(x)), & \sigma(x) &\geq 0, \\ \sigma(x) &= W(-Y(x)), & \sigma(x) &< 0. \end{aligned} \tag{34}$$

The Lambert-$W$ function encompasses two real-valued branches: $W_0(x)$ and $W_{-1}(x)$. Depending on the domain and range of these branches, the solution to Eq. (34) is derived as:

$$\sigma(x) = \begin{cases} W_{-1}(-Y(x)), & \sigma < -1, \\ W_0(-Y(x)), & -1 \leq \sigma < 0, \\ W_0(Y(x)), & \sigma \geq 0. \end{cases} \tag{35}$$

The entry-dominated solution ($\sigma_\alpha$) and exit-dominated solution ($\sigma_\beta$) can be obtained to align with the left and right boundary densities, respectively. These solutions can then be converted back to yield the solutions $\rho_\alpha$ and $\rho_\beta$ in terms of the Lambert-W function, as follows:

$$\begin{aligned} \rho_\alpha(x) &= \frac{1}{2}\left(\frac{K^*+1}{K^*-1}\left(W_{-1}(-Y_\alpha(x))+1\right)+1\right), \\ \rho_\beta(x) &= \frac{1}{2}\left(\frac{K^*+1}{K^*-1}\left(\sigma_\beta(x)+1\right)+1\right), \end{aligned} \tag{36}$$

where $\sigma_\beta(x)$ is given as:

$$\sigma_\beta(x) = \begin{cases} W_0(Y_\beta(x)), & 0 \leq \beta^* \leq 1-\rho_l, \\ 0, & \beta^* = 1-\rho_l, \\ W_0(-Y_\beta(x)), & 1-\rho_l \leq \beta^* \leq \frac{1}{2}. \end{cases} \tag{37}$$

Note that similar to the TASEP, the density solution $\rho_\alpha$, associated with the low-density regime, remains stable for $\alpha < 1/2$, while the solution corresponding to the high-density regime, $\rho_\beta$, is stable for $\beta^* \leq 1/2$.

Similar to the scenario with $K = 1$, we now derive a comprehensive solution for the density profile by considering various feasible combinations of the solutions $\rho_\alpha$ and $\rho_\beta$. In the parameter range where $\alpha, \beta^* \leq \frac{1}{2}$, different solutions emerge depending on whether $1-\beta^*$ surpasses, falls short of, or equals $\rho_l$. These solutions converge towards the Langmuir isotherm within the bulk while satisfying both boundary conditions [24]. When $\beta^* = 1-\rho_l$, a flat profile of $\rho_\beta$ is obtained, aligning with the Langmuir isotherm value $\rho_l$. Within this range, a domain wall emerges, characterized by a density expressed through a combination of $\rho_\alpha(x)$ and $\rho_\beta(x)$, given by:

$$\rho(x) = \begin{cases} \rho_\alpha(x), & x \leq x_w, \\ \rho_\beta(x), & x > x_w, \end{cases} \tag{38}$$

where $x_w$ is the position of the domain wall that can be determined utilizing the condition $\rho_\alpha(x_w) = 1 - \rho_\beta(x_w)$. The height of the domain wall $\Delta$ is given by $\rho_\beta(x_w) - \rho_\alpha(x_w)$. If $0 < x_w < 1$, a region consisting of a shock (S) phase is formed. If $x_w > 1$ then the lattice is in the low-density regime whose bulk is characterized by the density profile $\rho_\alpha(x)$ with a boundary layer on the right end. If $x_w < 0$ then the lattice in a high-density regime whose density profile is characterized by $\rho_\beta(x)$ with a boundary layer on the left end. In the left-region phases ($\alpha < 1/2, \beta^* < 1/2$), the phase boundaries extend for $\beta^* > 1/2$, remaining independent of the exit rate $\beta$ and aligned parallel to the $\beta$−axis. When $\alpha = 1/2$, the system transitions into the High-Density (HD) phase, where the bulk profile fails to match the entry rate, resulting in a boundary layer at the left end. Further increases in $\alpha$ primarily affect this boundary layer at the left end. However, an increase in $\beta^*$ beyond $1/2$ introduces a boundary layer at the right end. Consequently, the HD phase for $\beta^* \geq 1/2$ stands distinct from the HD phase for $\beta^* < 1/2$. In the bulk, the density profile remains unaffected by the entrance and exit rates, $\alpha$ and $\beta$, at the left and right boundaries. It is characterized by the extremal solution $W_0(-Y_{\beta=1/2})$ and is termed the "High-Density Meissner (HD$_M$)" phase. Hence, we deduce that

a maximum of four possible stationary phases can occur in the phase diagram for $K^* > 1$ that are LD, HD, S, and $HD_M$ phase.

Obtaining a generalized analytical stationary-state solution for Eq. (11a) poses a significant challenge due to the presence of complex features, including dynamic disorder and non-conserving particle dynamics within our system. Therefore, numerical techniques serve as a viable alternative for solving it, and this approach has been widely adopted in the literature to approximate solutions for such intricate systems. The time derivative term is retained in the system, and the steady-state solution is captured using significantly large time steps. The model equation is discretized using a finite difference scheme, employing the first-order forward difference formula for the time derivative and the second-order central difference formula for space derivatives; refer to B for further details.

## 6  Results & discussion

In previous sections, the analysis has been conducted with respect to the parameter $K^*$, which was introduced to simplify and solve Eq. (11a), which involves two parameters that are obstruction factor $z$ and the binding constant $K$. Both $z$ and $K$ are of great relevance as defect dynamics are controlled by $z$, and it quantify the hindrance caused by the defects to the particle movement, whereas $K$ is responsible for particle dynamics and signifies the ratio of particle attachment with respect to particle detachment. Therefore, we now again introduce them to investigate the effect of each of these parameters individually and compare the results with the existing literature. We begin with the analytical construction of the phase diagrams within the $\alpha - \beta$ plane utilizing the theoretical results obtained in the last two sections in order to study the effect of $z$, $K$, and $\Omega_d$ on the system's stationary characteristics. We conduct numerical Monte Carlo simulations employing the Gillespie algorithm with a random sequential update rule to verify our theoretical conclusions; please refer to Sec. A in the appendix for further details. Note that if the Markovian dynamics of the Langmuir Kinetics model (adsorption or desorption processes) is visualized as a graph where different configurations of the process represent nodes, and each allowed transition is a directed edge with a rate based on the process rules, then if we describe the equilibrium distribution of LK using Boltzmann weight, then the effective Hamiltonian is evaluated as $H = -k_B T \sum_2^{L-1} \sigma_i \ln K$, where the "energy" term is expressed as the sum over the product of the logarithm of the binding constant $K$ with the occupancy number of $i^{th}$ site (where $k_B$ is the Boltzmann weight, and $T$ is the temperature of the system). In this distribution, the scenario where $K = 1$ presents intriguing topological consequences, where the edges in the graph structure for Langmuir Kinetics (LK) lose their directionality [24]. Hence, it is expected that $K$ will significantly affect the topology of the phase diagram in the $\alpha - \beta$ plane. To investigate the individual impact of each of these parameters on the system's stationary properties, we initially fix the parameter $K$ and vary the rest. It must be noted that as we discuss the role of $K$ and $z$ individually, unlike the ref. [24], the transformations: $\rho(x) \leftrightarrow 1 - \rho(1-x)$, $w_a \leftrightarrow w_d$ no longer implies $K \leftrightarrow 1/K$. Therefore, the analysis is done for every choice of $K$, namely, $K = 1$, $K > 1$, and $K < 1$, where we further study the impact of $z$ and $\Omega_d$ on the steady-state features in each of these cases. Furthermore, the phase diagrams are developed in each scenario specifically for faster defect dynamics ($k^+, k^- \gtrsim 1$), as the naive mean-field approximation aligns closely with the Monte Carlo results within this parameter range [42]. We initially constructed the phase diagrams using the analytical expressions of the phase separation lines. Subsequently, to verify the proximity of these lines, we conduct Monte Carlo simulations at points near these lines, with detailed information provided in A. Note that the phase boundaries determined through simulation depend on the magnitude of the defect binding/unbinding rates. The lower rates result in deviations from

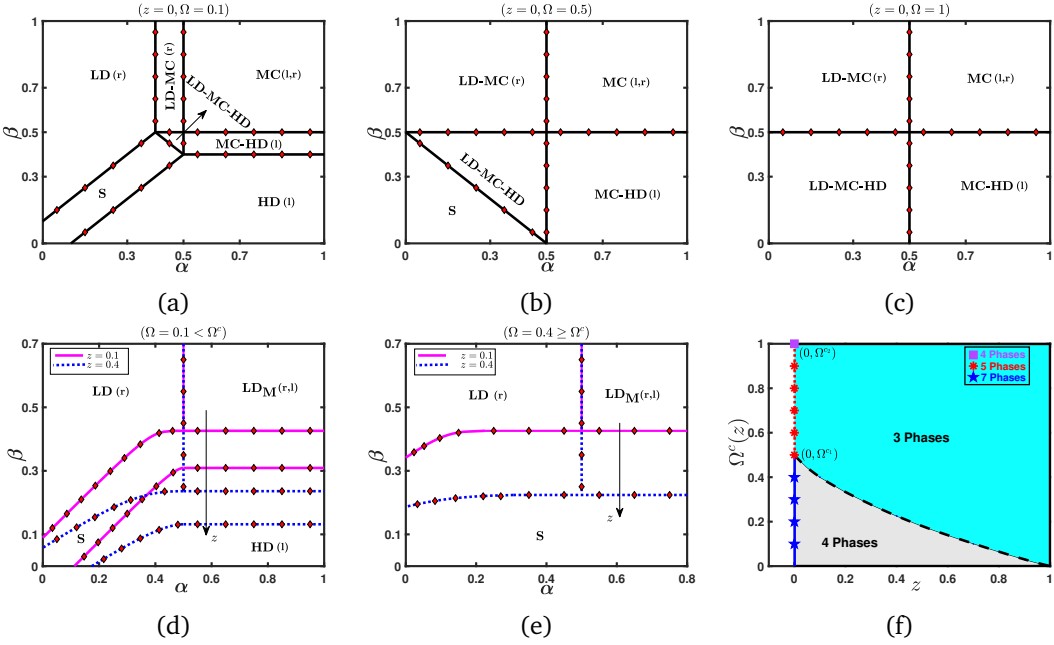

Figure 2: (a)-(e) demonstrates how the phase diagram is influenced by $z$ and $\Omega$ when $K = 1$. Solid and dashed lines represent theoretical predictions derived from mean-field theory, while Monte Carlo simulation results are depicted with diamonds. The presence of boundary layers at the left or right end of the system is highlighted by "(l)" and "(r)" respectively. In (f), the plot illustrates $\Omega^c$ as a function of $z$, with different symbols (stars, circles, and squares) distinguishing the two critical values $\Omega^{c_1}$ and $\Omega^{c_2}$ obtained for $z = 0$.

mean-field predictions due to system correlations, while faster defect dynamics align more closely with theoretically obtained results. In our model, we have considered these rates to be equal to or strictly greater than 1. Moreover, the phase boundaries determined through simulations are calculated with an estimated error of less than 2%, and the same is being taken care of by the size of the markers representing the Monte Carlo simulations.

## 6.1 System behavior for $K = 1$

In this context, the mathematical analysis is streamlined due to the equivalence of the attachment and detachment rates, denoted by $\Omega_a = \Omega_d \equiv \Omega$. Subsequently, we delve into an examination of the phase diagram's structure within the $\alpha - \beta$ parameter space, exploring its variations across different values of $z$ and $\Omega$. For $K = 1$, $K^*$ is a monotonically decreasing function of $z$ and it assumes values $K^* \leq 1$ for $z \in [0, 1)$. To assess the influence of the obstruction factor, we generate the phase diagrams for various choices of $\Omega$. Additionally, the effect of $z$ is examined for a fixed $\Omega$ by varying $z$. The upper and lower panels of FIG. 2 depict the phase diagrams corresponding to $z = 0$ and $z \neq 0$, respectively.

For different values of $\Omega$, we retrieve exactly the same phase diagrams obtained in ref. [24] in the limit $z \rightarrow 0$. We reproduce them here for the sake of comparison and analyzing the effect of $z$. For $\Omega < \Omega^{c_1} = 0.5$, a comparatively richer phase diagram exhibiting seven stationary phases is observed, as shown in FIG. 2 (a). An increase in $\Omega$ till the critical value $\Omega^{c_1}$ doesn't produce any topological changes in the phase diagram except the shifting of the phase boundaries. The boundary between the LD and LD-MC phases shifts leftward, while the boundary between the HD and MC-HD phases shifts downward. This leads to an enlargement of the LD-MC, HD-MC, and LD-MC-HD phases and a reduction of the LD, HD, and S phases,

while the MC phase remains unaffected. Once $\Omega$ reaches the critical value $\Omega^{c_1}$, the LD and HD phases completely disappear from the phase diagram, and it now consists of five stationary phases only, see FIG. 2 (b). Further increasing $\Omega > \Omega^{c_1}$ only affects the region $\alpha, \beta < 0.5$, where the S phase shrinks and the LD-MC-HD expands whereas the LD-MC and MC-HD phases remain intact. Ultimately, at $\Omega = \Omega^{c_2} = 1$, the S phase vanishes entirely, rendering the phase diagram greatly simplified, with only four phases remaining, as depicted in FIG. 2 (c).

Now, we investigate the effect of the obstruction factor on the phase diagram for different choices of attachment-detachment rates. As soon as some obstruction is introduced in the lattice, the topology of the phase diagram changes drastically and becomes much simpler, consisting of four phases, see FIG. 2 (d) in comparison to the phase diagram obtained for zero obstruction factor, see FIG. 2 (a). For $\Omega < \Omega^c(z)$, the phase diagram consists of LD, S and HD phases along with the emergence of a $LD_M$ phase; see FIG. 2 (d) corresponding to $\Omega = 0.1$. Further, increasing the obstruction factor on the lattice while fixing $\Omega$ results in an expansion of the LD and $LD_M$ phases, whereas the region consisting of the S and HD phases shrinks. This can be explained as follows: an escalation in the obstruction factor intensifies the impedance to particle movement throughout the lattice, consequently enlarging the domain encompassing both the LD phase and the $LD_M$ phases. Likewise, augmenting $\Omega$ enlarges the area encompassing the LD and S phases while the HD phase diminishes. In instances where $\Omega \geq \Omega^c(z)$, we note the total absence of the HD phase, resulting in a phase diagram comprising only three phases: LD, $LD_M$, and S phases, as depicted in FIG. 2 (e). In this case, the effect of increasing $z$ remains the same. The FIG. 2 (f) shows the graph of the $\Omega^c$, which is a monotonically decreasing function of $z$. The graph demonstrates that for $z > 0$, there is only one critical value of $\Omega$, beyond which the number of stationary phases appearing in the phase diagram decreases from four to three. Nevertheless, when $z = 0$, two critical values exist: $\Omega^{c_1} = 0.5$ and $\Omega^{c_2} = 1$. For $\Omega \geq \Omega^{c_1}$, the number of stationary phases decreases from seven to five, while for $\Omega \geq \Omega^{c_2}$, it decreases from five to four.

The phase diagram's structure differs significantly when considering equal attachment-detachment rates and a non-zero obstruction factor compared to the results obtained in ref. [24] (refer to the top and bottom panels of FIG. 2). Clearly, the presence of defects in the proposed model for the equal attachment-detachment rate of particles has made the phase diagram much simpler, which can possess at most four stationary phases depending upon the choice of $\Omega$ and $z$.

## 6.2 System behavior for $K > 1$

In general, one would anticipate $K \neq 1$ because the case $K = 1$ requires a specific adjustment between the attachment and detachment rates. Therefore, without loss of generality, we first discuss the case $K > 1$ and try to understand the effect of $z$ and $\Omega_d$ on the stationary state features of the system. Analogous to the previous case, we first examine the influence of the obstruction factor by delineating the phase diagram for various selections of $\Omega_d$. In contrast to the prior scenario, in this case, the parameter $K^*$ varies depending on both $K$ and $z$. We have three different cases corresponding to the range of $z$ according to which $K^*$ is either $> 1$ or $= 1$ or $< 1$. The panels at the top, middle, and bottom of FIG. 3 depict the phase diagrams corresponding to $z$ values within the ranges $\left[0, \frac{K-1}{K}\right)$, $z = \frac{K-1}{K}$, and $\left(\frac{K-1}{K}, 1\right)$, respectively.

In the limit $z \to 0$, the phase diagram for $\Omega_d < \Omega_d^c(z, K)$ consists of four stationary phases: LD, S, HD, and $HD_M$, see FIG. 3 (a). It validates the findings of the ref. [24] corresponding to $K = 3$ and $\Omega_d = 0.1$. As soon as some obstruction is introduced in the lattice, i.e., for $z \in \left(0, \frac{K-1}{K}\right)$, the phase boundary between the LD and the S phase as well as the one between the $HD_M$ and the S phase shifts towards the right resulting in shrinkage in the region consisting of $HD_M$ and HD phase whereas an expansion of the region consisting of LD and the S phase. Unlike the scenario with $K = 1$, the inclusion of the obstruction doesn't induce significant

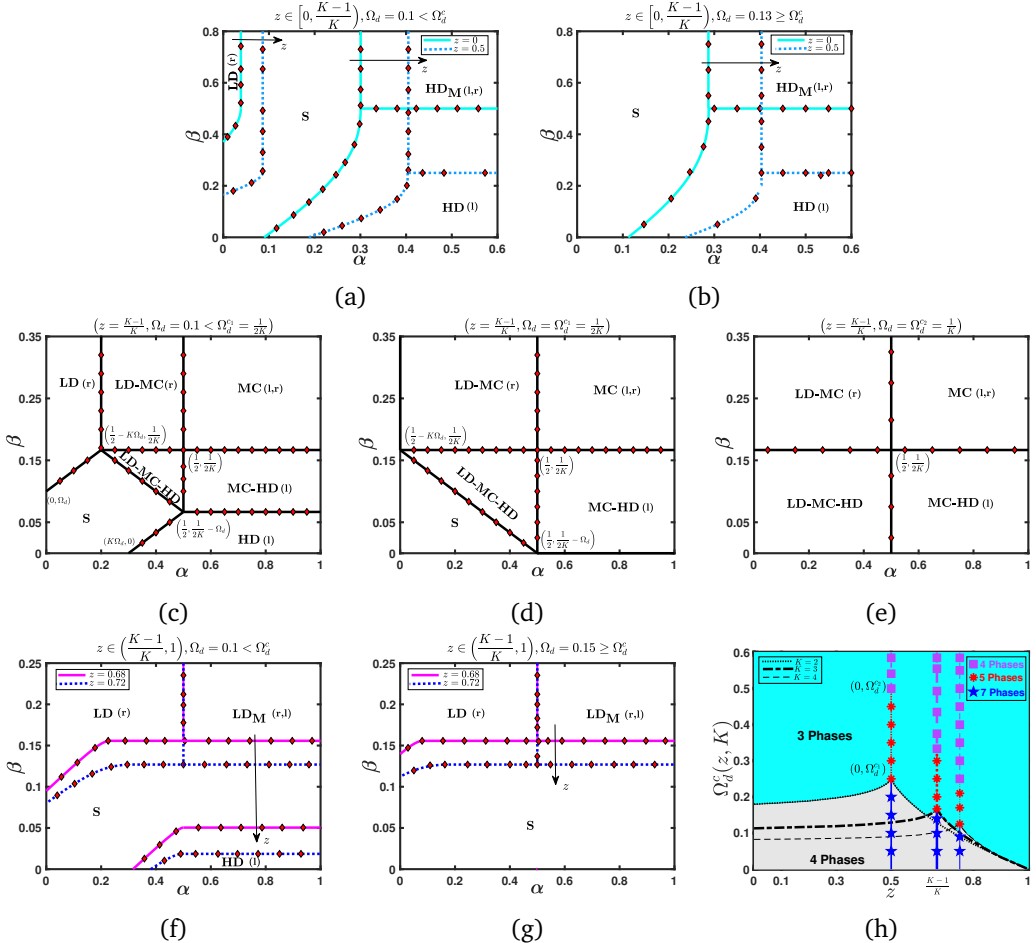

Figure 3: (a)-(g) shows the effect of $z$ and $\Omega_d$ on the phase diagram for $K = 3$. Solid and dashed lines denote the theoretical outcomes through mean-field theory, whereas diamonds denote the Monte Carlo simulation results. In (h), the plot shows $\Omega_d^c$ as a function of $z$ and $K$. Here, the symbol star, circle, and square distinguish the two critical values $\Omega^{c_1}$ and $\Omega^{c_2}$ obtained corresponding to $z = \frac{K-1}{K}$.

topological alterations in the phase diagram, except for expansions and contractions in the regions encompassing stationary phases. When $\Omega_d \geq \Omega_d^c(z, K)$, the boundary separating the LD and S phases shifts leftward, leading to the total absence of the LD phase. Consequently, the phase diagram comprises only three stationary phases, as depicted in FIG. 3 (b). The impact of varying $z$ in $(0, \frac{K-1}{K})$ remains the same for this choice of $\Omega_d$.

Once the obstruction factor reaches $\frac{K-1}{K}$, as illustrated in FIG. 3 (c), FIG. 3 (d), and FIG. 3 (e), the phase diagram undergoes notable topological changes. The phase diagram becomes more intricate and diverse for values of $z$ smaller than $\frac{K-1}{K}$. It showcases seven stationary phases when $\Omega_d < \Omega_d^{c_1}(K) = \frac{1}{2K}$, as depicted in FIG. 3 (c). The average density profiles for these seven stationary phases have been obtained in FIG. 4. As $\Omega_d$ increases till the critical value $\Omega_d^{c_1}(K)$, the phase boundary separating LD and LD-MC phases shifts to the left, whereas the phase boundary between the HD and MC-HD phases shifts downward, leading to an enlargement of LD-MC, HD-MC, and LD-MC-HD phases, and a contraction of LD, HD, and S phases, while the MC phase remains unaffected. For $\Omega_d = \Omega_d^{c_1}(K)$, the LD and HD phases completely disappear from the phase diagram, and now it consists of only five stationary phases; see FIG. 3 (d). As $\Omega_d$ increases in the range $(\Omega_d^{c_1}(K), \Omega_d^{c_2}(K) = \frac{1}{K})$, the phase diagram is only

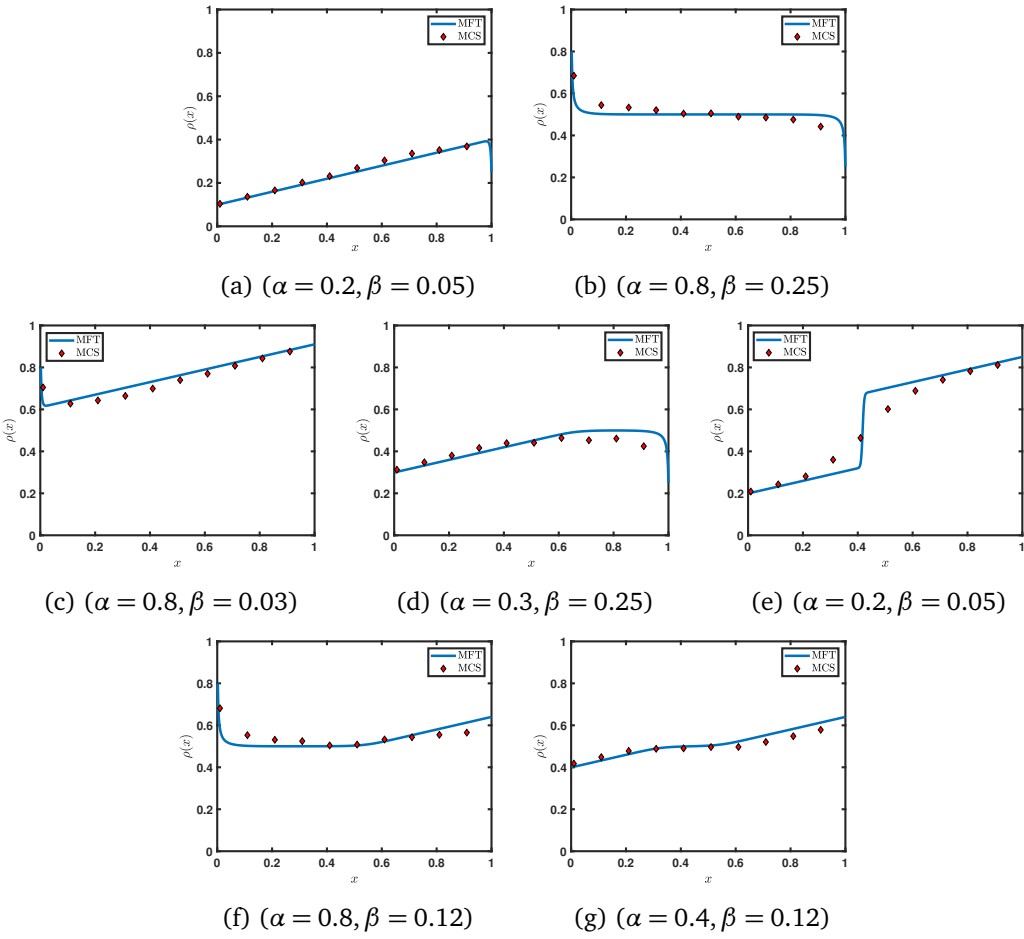

Figure 4: Average density profiles: (a) LD, (b) MC, (c) HD, (d) LD-MC, (e) LD-HD, (f) MC-HD, and (g) LD-MC-HD phases. Mean-field solutions (solid curves) contrasted with Monte Carlo simulations (markers). Parameters: $K = 3$, $\Omega_d = 0.1$, $z = \frac{K-1}{K}$. Sub-captions detail $(\alpha, \beta)$ configurations.

affected in the region $\alpha, \frac{\beta}{1-z} < 0.5$, where the S phase shrinks and the LD-MC-HD expands, whereas the phases LD-MC, MC-HD, and MC remain intact. Finally, when $\Omega$ equals $\Omega_d^{c_2}(K)$, the S phase vanishes entirely, rendering a simpler phase diagram exhibiting just four phases; refer to FIG. 3 (e).

Now, we discuss the case $z > \frac{K-1}{K}$. The phase diagram again becomes topologically simpler, as shown in FIG. 3 (f) and FIG. 3 (g). For $\Omega_d < \Omega_d^c(z, K)$, the phase diagram showcases four stationary phases: HD, S, LD, and $LD_M$, as shown in FIG 3 (f) corresponding to $\Omega = 0.1$. For a further increase in $z$ in the range $\left(\frac{K-1}{K}, 1\right)$, the phase boundary separating the S phase from LD as well as the $LD_M$ phase shifts downwards, this leads to an enlargement of the LD and $LD_M$ phases, while the S and HD phases diminish. For $\Omega_d \geq \Omega_d^c(z, K)$, the phase boundary between the HD and S phase shifts downwards, causing the complete disappearance of the HD phase, and the phase diagram exhibits only three stationary phases; see FIG. 3 (g). The influence of varying $z$ in $\left(\frac{K-1}{K}, 1\right)$ remains the same for this choice of $\Omega_d$.

In comparison to the case $K = 1$, FIG. 3 (h) shows the critical values of $\Omega_d^c$ which is a function of $z$ as well as $K$. For a fixed $K$, $\Omega_d^c(z, k)$ is a non-monotonic function that monotonically increases for $z < \frac{K-1}{K}$, whereas it monotonically decreases for $z > \frac{K-1}{K}$ and attains its maximum value at $z = \frac{K-1}{K}$. Meanwhile, for a fixed value of $z$, it is a monotonically decreasing function of $K$. Clearly, for $z \neq \frac{K-1}{K}$, there exists only one critical value of $\Omega_d$, beyond which

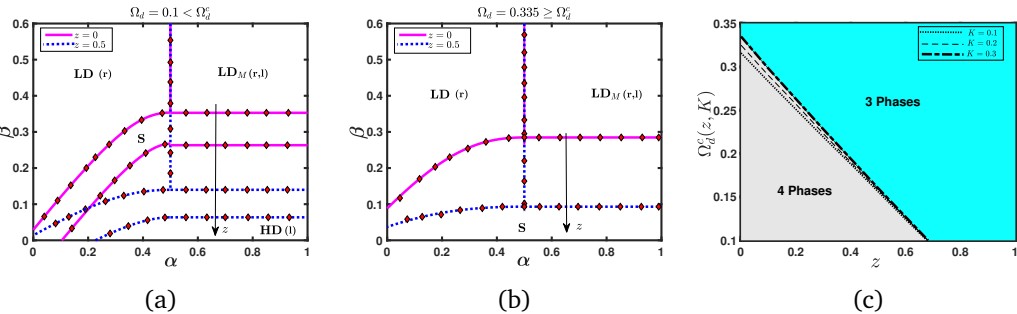

Figure 5: (a)-(b) shows the effect of $z$ and $\Omega_d$ on the phase diagram for $K = 0.3$. Solid and dashed lines denote the theoretical outcomes through mean-field theory, whereas diamonds denote the Monte Carlo simulation results. In (c), the graph depicts $\Omega_d^c$ varying with $z$ and $K$.

the number of stationary phases changes from four to three. However, when $z = \frac{K-1}{K}$, two critical values emerge: $\Omega_d^{c_1} = \frac{1}{2K}$ and $\Omega_d^{c_2} = \frac{1}{K}$. In this scenario, if $\Omega$ exceeds $\Omega_d^{c_1}$, the count of stationary phases decreases from seven to five. Likewise, when $\Omega$ surpasses $\Omega_d^{c_2}$, the number of stationary phases decreases from five to four. It must be noted that $z^c = \frac{K-1}{K}$ is the general critical value of the obstruction factor for which the phase diagram's structure becomes intricate, featuring a maximum of seven stationary phases.

For an attachment rate larger than the detachment rate, the topology of the proposed model's phase diagrams remains the same compared to the ref. [24] for $z < z_c$ except for expansion or shrinkage of some phase regions (see top panel of FIG.3). But for $z \geq z_c$, the topology changes significantly in comparison to the ref. [24] (see middle and bottom panels of FIG.3). Depending on the effective binding constant $K^*$, the phase diagram of the proposed model can exhibit several stationary phases, including LD-MC-HD, LD-MC, MC-HD, MC, and $LD_M$.

## 6.3 System behavior for $K < 1$

Due to the defects considered in the proposed model, the particle-hole symmetry is violated with respect to $K$, as discussed at the beginning of this section. Therefore, the case $K < 1$ needs to be discussed separately. Similar to the previous case, we first establish the phase diagram for various $\Omega_d$ choices and analyze the influence of the obstruction factor. The parameter $K^*$ is a function of both $K$ and $z$ in this case as well, and $K^* < 1$ for any possible combination of $z$ and $K < 1$.

As $z$ approaches zero, the phase diagram exhibits four stationary phases: high density (HD), shock (S), low density (LD), and $LD_M$, for $\Omega_d = 0.1 < \Omega_d^c(z)$, as depicted in FIG 5 (a). It validates the findings of the ref. [24] corresponding to $K = 0.3$ and $\Omega_d = 0.1$. When some obstruction is introduced to the lattice, the phase boundary that separates the S phase from the LD as well as $LD_M$ phases shifts downward, expanding the LD and $LD_M$ phases while contracting the S and HD phases, as shown in FIG. 5 (a). However, when $\Omega_d = 0.335 \geq \Omega_d^c(z)$, the high-density (HD) phase is entirely absent, resulting in a phase diagram with only three stationary phases, as illustrated in FIG. 5(b). The influence of $z$ within the $(0, 1)$ range remains consistent for this particular value of $\Omega_d$.

The FIG. 5 (c) shows the graph of the $\Omega_d^c$ which is a linear as well as monotonically decreasing function of $z$ and $K$. The graph demonstrates that for $z \in [0, 1)$ there exists only one critical value of $\Omega_d$, beyond which the number of stationary phases appearing in the phase diagram reduces from four to three.

For an attachment rate smaller than the detachment rate, the obstruction factor does not change much topology of the phase diagram in comparison to the ref. [24] except for the shrinkage and expansion of the phase region. Now, we briefly revisit the link mentioned in Section 2 between the proposed model and the investigation presented in Ref. [45] to showcase several distinctions in their stationary state results. Firstly, the impact of parameters $\rho_d$ and $p_d$, responsible for obstructions caused by defects, on the stationary state characteristics of the system, encapsulated through a single parameter $z$. Secondly, in the scenario of equal attachment-detachment rates and non-zero obstruction, the proposed model exhibits a maximum of four stationary phases in its phase diagrams, while in [45] the phase diagram consists of seven stationary phases. For this case, the phase diagram also includes a low-density Meissner ($HD_M$) phase, which was not observed in the [45]. Moreover, the topology of these phase diagrams differs significantly from our observations in [45]. In the case of disparate attachment-detachment rates and non-zero obstruction, the phase diagram within our proposed study can feature up to seven stationary phases in the phase diagram, while in [45], the system can exhibit a maximum of four stationary phases. The configurations of these phase diagrams exhibit variations compared to the findings in [45], contingent upon the selection of attachment-detachment rates and the obstruction factor. This discrepancy can be elucidated by considering the significance of boundary densities in an open system, as they strongly influence the stationary properties and phase diagrams. The inclusion of the effects of defects binding/unbinding at the boundaries, which is absent in ref. [45], is a probable reason for this distinction. Lastly, unlike the model proposed in [45], our system's stationary-state results are obtained analytically, providing a comprehensive characterization of the influence of all parameters.

## 7  Shock analysis & finite-size effect

One distinctive aspect of the proposed model is the emergence of the localized shock (S) phase, where the shock position remains constant over time. Typically, a qualitative examination of shock dynamics can be straightforwardly conducted using the continuity (or hydrodynamic) equation, which is expressed as:

$$\frac{\partial \rho}{\partial t'} + \frac{\partial J}{\partial x} = \omega_d L(K^* - (1 + K^*)\rho). \tag{39}$$

In this context, the flow-density relation, denoted by $J = (1-z)\rho(1-\rho)$, is well-established, allowing for the analysis of the equation above. However, the description provided by the first-order differential equation (39) becomes invalid as soon as a discontinuity arises between the densities $\rho_\alpha$ and $\rho_\beta$ and at the intersection points of the characteristic lines corresponding to (39). This discontinuity propagates at a speed $v = \beta^* - \alpha$, determined by the balance of mass current. To establish the formation of a shock, the discontinuity must reach a position where the mass current through it is zero, thus ensuring the shock remains stationary, indicating that $v$ must be zero.

Progressing further involves analyzing how the obstruction factor influences the shock profiles. A comprehensive shock profile spanning the entire system can be derived by aligning the boundary densities $\rho_\alpha$ and $\rho_\beta$ at the location of the shock, which needs to be identified. For $K = 1$, the precise formulae for the shock position ($x_w$) and its height ($\Delta$) are provided as follows:

$$x_w = \frac{\beta - \alpha(1-z) + \Omega_d}{2\Omega_d}, \quad \text{and} \quad \Delta = 1 - (\alpha + \beta) - \frac{\Omega_d}{1-z}. \tag{40}$$

Evidently, the shock's position is consistently influenced by $z$, increasing as $z$ increases, while its height shows the opposite trend, decreasing as $z$ increases. Although obtaining explicit

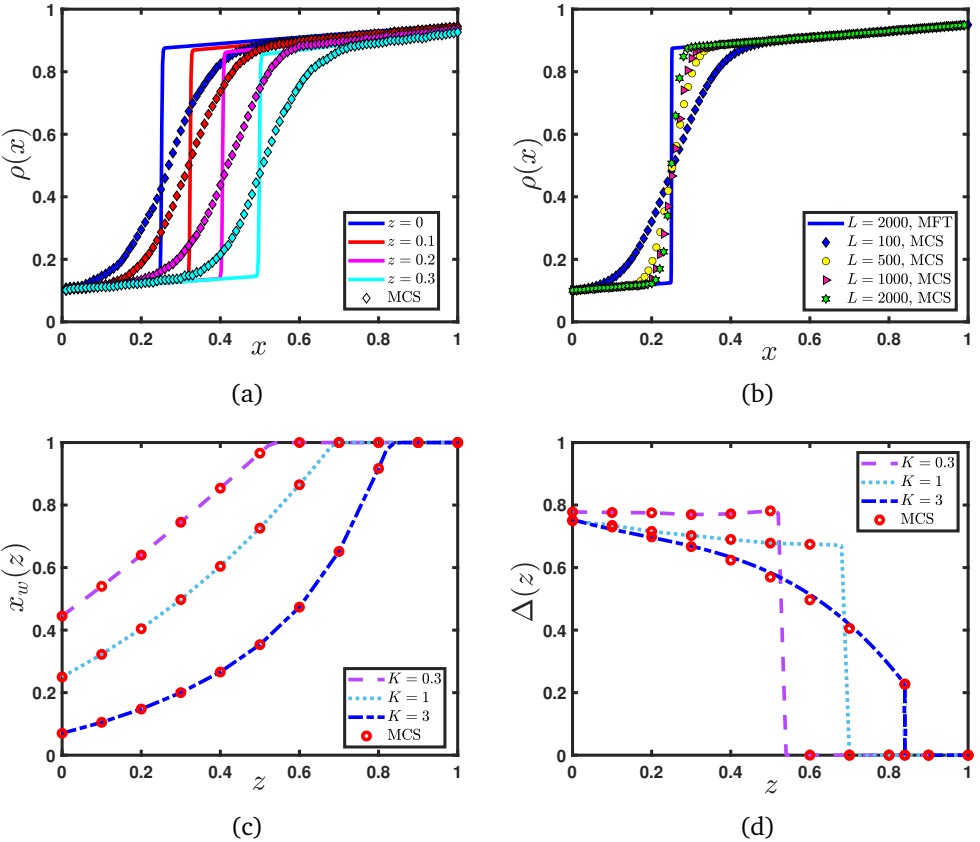

Figure 6: (a) Examining shock profiles under varying $z$ for $\alpha = 0.1$, $\beta = 0.05$, $\Omega_d = 0.1$, and $K = 1$. (b) Finite-size effects on shock profiles at $z = 0$. Investigating (c) shock displacement and (d) shock amplitude across $z$.

expressions for $x_w$ and $\Delta$ for $K^* \neq 1$ remains challenging, their corresponding Monte Carlo results are depicted in FIG. 6 (c) and 6 (d), with fitted curves confirming their dependency on $z$. In FIG. 6 (a), for fixed values of $\alpha$, $\beta$, and $K$, it's evident that the shock profile shifts from the left to the right boundaries with increasing obstruction on the lattice. This shift occurs because as z increases, particles encounter more obstructions from defects, decreasing particle density. Consequently, the HD phase's elimination and the LD phase's expansion are observed. These findings are consistent with the phase diagrams discussed in the preceding section.

Finite-size effects for finite $L$ have been accounted for by incorporating second-order terms in the mean-field description. Discrepancies between the second-order mean field and the Monte Carlo results arise from shock fluctuations, which are inaccurately captured by mean-field theory and require separate treatment. Nonetheless, it's noteworthy that the shock is indeed localized, and its width grows sub-extensively, indicating sharpness as $L \to \infty$, as shown in FIG 6 (b).

# 8 Conclusion

We've extensively examined stochastic transportation within a one-dimensional system, incorporating dynamic disorder in a totally asymmetric simple exclusion process alongside Langmuir kinetics dynamics. The dynamic defects represent disorder that stochastically binds/unbinds throughout the lattice and hinders particle movement. The particle movement

has been subject to these dynamic defects and hops with an affected hopping rate $p_d$. The model is inspired by the imbalance in the transcription of genes due to obstruction, but the model is generic and can be utilized to imitate any other non-equilibrium stochastic transport phenomena where dynamic defects are present.

To explore how dynamic defects influence the stationary-state characteristics of the system, we derive master equations in the thermodynamic limit under the framework of the continuum mean-field approximation. Moreover, we introduce a parameter termed the obstruction factor ($z$), which amalgamates the impact of defect density on the lattice ($\rho_d$) and the affected hopping rate ($p_d$) on the system's stationary properties. In addition, we define an effective binding constant that incorporates the effect of obstruction on the binding constant. The system dynamics are controlled by the entry rate ($\alpha$) and exit rate ($\beta$), and the other three important controlling parameters are $z$, binding constant ($K$), and total detachment rate $\Omega_d$. The explicit expression of analytical solutions for the density profile and phase boundaries are obtained for $K = 1/(1-z)$ whereas, for the rest of the values of $K$, the stationary state solution has been implicitly expressed in the form of Lambert $W$ function.

The theoretical solution has enabled us to delineate and extensively analyze the topology of the phase diagram. Since the proposed model doesn't obey the particle-hole symmetry, the analysis is performed for three distinct choices of binding constant, $K = 1, > 1$ and $< 1$. In each case, the $z$ and $\Omega_d$ effects have been studied on the phase diagram. At the critical value of the obstruction factor ($z^c$), the topology of the phase diagram changes significantly. At $z = z^c$, the phase diagram displays a richer structure consisting of either seven or five, or four stationary phases depending upon the value of $\Omega_d$. Whereas for $z \neq z^c$, the phase diagram consists of either four or three stationary phases depending on $\Omega_d$. In this case, with an increase in the magnitude of the obstruction factor, the LD or $LD_M$ phases expand, whereas the $S$ phase and the HD phase shrink. Furthermore, the impact of $\Omega_d$ on the phase diagrams is explored, revealing that an escalation in $\Omega_d$ diminishes the number of phases within the system. For $z \neq z^c$, there exists a unique critical value $\Omega_d^c$ about which the number of stationary phases changes from three to four. While for $z = z^c$, there exist two critical values of $\Omega_d^{c_1}$ and $\Omega_d^{c_2}$ such that about $\Omega_d^{c_1}$, the number of phases changes from seven to five whereas about $\Omega_d^{c_2}$, the number of phases changes from five to four. The variation in the number of stationary phases with respect to the obstruction parameter $z$ can be understood as follows: with the increase in the obstruction factor, the dynamic defects increasingly hinder the particle flux in the bulk, effectively making the bulk dynamics more rate-limiting. Despite the bulk becoming rate-limiting, the boundary dynamics ($\alpha$ and $\beta$) still play a significant role in determining stationary phases. Consequently, the interplay between bulk obstructions and boundary conditions influences the number of distinct stationary phases that the system can possess. This behavior highlights the critical role of the obstruction parameter in dictating the system's overall phase structure. Further, we examine the impact of the obstruction factor on the height (and position) of the de-localized shock, which is a monotonically decreasing (increasing) function of $z$. Finally, we conclude that the proposed theoretical work aimed to simulate dynamic aspects of potential defective cellular and vehicular transport processes and to provide light on stationary qualities. The proposed study can be utilized to understand the role of the disorder in the form of defects on the stationary properties of the stochastic transport systems. Examples of such systems include the biological process of gene transcription, where DNA binding proteins and the low concentration of tRNA act as a disorder [25], transport processes along the microtubule where processive molecular motors switch between directed and diffusive motion [21] etc. The study can be expanded to include additional realistic aspects relevant to various physical and biological systems.

## Acknowledgments

**Funding information**   The first author thanks the Council of Scientific and Industrial Research (CSIR), India, for financial support under File No:09/1005(0028)/2019-EMR-I.

## A   Monte Carlo simulations

For simulations, we utilize a Monte Carlo algorithm (Gillespie Algorithm) with a random sequential update rule [46]. A random site is selected and updated at each step according to events like particle hopping attempts, attachment or detachment, and defect binding or unbinding, chosen with rates outlined in Sec. 2. Time increments follow an exponentially distributed random pattern. The simulations are conducted for $10^8$ time steps considering the lattice size $L = 500$. The initial 5% of time steps are discarded to establish a steady state, and the average particle density is calculated over an interval of $10L$. Phase boundaries are determined with an estimated error of less than 2%, indicated by the marker sizes in the Monte Carlo simulations.

## B   Numerical scheme

To derive analytical solutions for the second-order partial differential equation given by Eq. (11a) can be challenging; hence, we present an alternative approach within the mean-field theory. Retaining the time derivative within the system, we obtain density solutions at a steady state in the limit as $t$ tends to $\infty$, where $t$ denotes the number of time steps. Employing the forward-in-time and central-in-space (FTCS) scheme, we derive the finite-difference equation as:

$$
\begin{aligned}
\rho_j^{i+1} = \rho_j^i + \Delta t' \Bigg( \left(1 - \rho_{d,j}^i(1 - p_d)\right) \Bigg( \frac{\epsilon}{2}\left( \frac{\rho_{j+1}^i - 2\rho_j^i + \rho_{j-1}^i}{\Delta x^2} \right) \\
+ \left( \frac{\rho_{j+1}^i - \rho_{j-1}^i}{2\Delta x} \right)(2\rho_j^i - 1) + \Omega_a(1 - \rho_j) \Bigg) - \Omega_d \rho_j \Bigg),
\end{aligned}
\tag{B.1}
$$

$$
\rho_{d,j}^{i+1} = \rho_{d,j}^i + \Delta t' \left( k^+(1 - \rho_{d,j}^i) - k^- \rho_{d,j}^i \right).
\tag{B.2}
$$

The symbols $\rho_j^i$ and $\rho_{d,j}^i$ represent the numerical approximation of particle density and defect density at the point $(x_j, t_i)$. Here, the spatial variable $\Delta x = 1/L$ and the temporal variable $\Delta t'$ adhere to the stability criterion of the finite-difference scheme mentioned above, $\Delta t'/\Delta x^2 \leq 1$. Similarly, Eq. (3) and Eq. (4) are employed to derive the finite-difference equations at the left and right boundaries as:

$$
\rho_1^{i+1} = \rho_1^i + \Delta t' \left( \left(1 - \rho_{d,1}^i(1 - p_d)\right)\left( \alpha(1 - \rho_1^i) - \rho_1^i(1 - \rho_2^i) \right) \right),
\tag{B.3}
$$

and

$$
\rho_L^{i+1} = \rho_L^i + \Delta t' \left( \left(1 - \rho_{d,L}^i(1 - p_d)\right)\left( \rho_{L-1}^i(1 - \rho_L^i) - \beta \rho_L^i \right) \right),
\tag{B.4}
$$

respectively.

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
