# Peer review of "Site-wise dynamic defects in a non-conserving exclusion process"

_SciPost Physics, doi:SciPost Phys. Core 8, 009 (2025)_

## Round 1 · Referee Report · Anonymous (Referee 1) · 2024-11-21

Report

The paper entitled 'Site-wise dynamic defects in a non-conserving exclusion process' by Nikhil Bhatia and Arvind Kumar Gupta is scientifically sound. It treats an original model of unidimensional transport with open boundaries and displaying defects. I recommend it for publication in SciPost.

Recommendation

Publish (meets expectations and criteria for this Journal)

---

## Round 1 · Referee Report · Anatoly Kolomeisky (Referee 2) · 2024-11-23

Strengths

This is a comprehensive theoretical/computational investigation of complex non-equilibrium multi-particle 1D transport systems. By combining mean0field analytical calculations and extensive Monte Carlo computer simulations, the authors specifically investigated a totally asymmetric exclusion model with dynamic disorder and reversible binding/unbinding of defect particles. The paper is well-written, and the results are well-explained. There is clear logic, and the conclusions are interesting. One important thing is that the authors emphasized how different their approach is from that of the existing one in the field. I like this work, and I think the results are impactful. There is a general interest in understanding non-equilibrium processes, and this work adds more insights into these important phenomena. The paper uncovers some new physics, and this is important.

Weaknesses

1) I would improve the quality of the figures. Some of them are hard to see. Symbols are too small and lines are barely visible. 2) I would add more physics explanations on why the number of stationary phases varies with the obstruction parameter. For example, one could say that increasing the parameter z makes the bulk dynamics more rate-limiting. But because other phases are also determined by entrance and exit, this should decrease the number of phases.

Report

To the best of my understanding, the paper meets the requirements of this journal.

Requested changes

1) Improve figures; 2) Add physics explanation as discussed above.

Recommendation

Publish (easily meets expectations and criteria for this Journal; among top 50%)

---

## Round 2 · Author Response

Dear Editor,

We sincerely thank you for reviewing our manuscript, "Site-wise dynamic defects in a non-conserving exclusion process." Your detailed comments and insightful suggestions have been incredibly valuable in helping us refine our work. We deeply appreciate your thoughtful feedback, which has enhanced the clarity and rigor of our study. Your contributions have significantly strengthened the overall quality of our manuscript, and we are grateful for your effort and expertise. All corrections in the manuscript are highlighted in red text.

Response to the comments of Reviewer 1

Reviewer #1: The paper entitled 'Site-wise dynamic defects in a non-conserving exclusion process' by Nikhil Bhatia and Arvind Kumar Gupta is scientifically sound. It treats an original model of unidimensional transport with open boundaries and displaying defects. I recommend it for publication in SciPost.

Response to the comments of Reviewer 2

Reviewer #2: This is a comprehensive theoretical/computational investigation of complex non-equilibrium multi-particle 1D transport systems. By combining mean0field analytical calculations and extensive Monte Carlo computer simulations, the authors specifically investigated a totally asymmetric exclusion model with dynamic disorder and reversible binding/unbinding of defect particles. The paper is well-written, and the results are well-explained. There is clear logic, and the conclusions are interesting. One important thing is that the authors emphasized how different their approach is from that of the existing one in the field. I like this work, and I think the results are impactful. There is a general interest in understanding non-equilibrium processes, and this work adds more insights into these important phenomena. The paper uncovers some new physics, and this is important.

Comment 1: I would improve the quality of the figures. Some of them are hard to see. Symbols are too small and lines are barely visible. Response: Thank you for your feedback. We have enhanced the sizes of the figures to make it more visible.

Comment 2: I would add more physics explanations on why the number of stationary phases varies with the obstruction parameter. For example, one could say that increasing the parameter z makes the bulk dynamics more rate-limiting. But because other phases are also determined by entrance and exit, this should decrease the number of phases. Response: Thank you for your insightful comment. We have incorporated a discussion in the conclusion section to explain the physics behind the variation in the number of stationary phases. The added text has been highlighted in red for your reference.

---

## Round 2 · List of Changes

1. We have enhanced the sizes of the figures to make it more visible as pointed out by the reviewer # 2.

  2. In response to the comment # 2 of reviewer # 2, we have incorporated a discussion in the conclusion section to explain the physics behind the variation in the number of stationary phases. The added text has been highlighted in red for your reference.

---

## Editorial Decision

published